# Multiagent Hierarchical Cognition Difference Policy for Multiagent Cooperation

**Huimu Wang** [1,2]**, Zhen Liu** [2,]*****, Jianqiang Yi** [1,2] **and Zhiqiang Pu** [1,2]

[1] School of Artificial Intelligence, University of Chinese Academy of Sciences, Beijing 100049, China; wanghuimu2018@ia.ac.cn (H.W.); jianqiang.yi@ia.ac.cn (J.Y.); zhiqiang.pu@ia.ac.cn (Z.P.)

[2] Institute of Automation, Chinese Academy of Sciences, Beijing 100190, China

***** Correspondence: liuzhen@ia.ac.cn

**Abstract:** Multiagent cooperation is one of the most attractive research fields in multiagent systems. There are many attempts made by researchers in this field to promote cooperation behavior. However, several issues still exist, such as complex interactions among different groups of agents, redundant communication contents of irrelevant agents, which prevents the learning and convergence of agent cooperation behaviors. To address the limitations above, a novel method called multiagent hierarchical cognition difference policy (MA-HCDP) is proposed in this paper. It includes a hierarchical group network (HGN), a cognition difference network (CDN), and a soft communication network (SCN). HGN is designed to distinguish different underlying information of diverse groups' observations (including friendly group, enemy group, and object group) and extract different high-dimensional state representations of different groups. CDN is designed based on a variational auto-encoder to allow each agent to choose its neighbors (communication targets) adaptively with its environment cognition difference. SCN is designed to handle the complex interactions among the agents with a soft attention mechanism. The results of simulations demonstrate the superior effectiveness of our method compared with existing methods.

**Keywords:** multiagent system; deep reinforcement learning; variational autoencoder; attention mechanism

## 1. Introduction

Grouping and effective communication are important methods to promote multiagent cooperative behavior. Agents in nature such as ants, social animals, and humans tend to cooperate and generate complex cooperative strategies by grouping and exchanging information. Naturally, the behaviors of grouping and exchanging information also apply to multiagent systems, especially in scenarios that require cooperation, such as smart grid control [1], resource management [2], and games [3,4].

Recently, deep reinforcement learning (DRL) has shown great potential in many domains, such as games [5,6] and robotics [7,8]. Inspired by the powerful perception and learning ability of DRL, researchers have made continuous attempts to apply DRL to multiagent reinforcement learning (MARL) to promote multiagent cooperative behaviors in environments with many agents [9–15]. Based on the common paradigm of centralized learning with decentralized execution, some MARL algorithms learn centralized critics for multiple agents and determine the decentralized action. However, when these methods are applied to environments with a large number of agents, they have their limitations. Some MARL algorithms [9,10] ignore different underlying influences brought by different groups of observation. Although some algorithms based on attention mechanism [16–18] partially consider the influences of groups, they do not take communication relationship among the agents into consideration. Furthermore, some algorithms [14,19,20] deal with the communication relationship between agents, but they do not consider the problem of

communication redundancy, making it unsuitable for environments with a large number of agents.

To address the limitations above, we propose a novel method called multiagent hierarchical cognition difference policy (MA-HCDP) to handle state representations of different group of agents and filter irrelevant agents to promote agents' cooperation behavior. In general, MA-HCDP can divide agents into different groups, filter out irrelevant agents in these groups, and handle the interactions among the remaining agents, in that order. Corresponding to the three functions is three networks in MA-HCDP, including a hierarchical group network (HGN), a cognition difference network (CDN), and a soft communication network (SCN). Specifically, HGN is responsible for extracting high-dimensional state representations of different groups including friend groups, enemy groups, and object groups. Then, the agents' understanding of the environment is extracted based on the group-level state representations obtained from SCN and posterior distributions of variational autoencoder (VAE) [21] in CDN. Next, the differences between distributions of the agents are calculated with Kullback–Leibler (KL) divergence [22]. If the difference is large, these agents are defined as irrelevant agents and filtered out. SCN is responsible for the weight distribution of the agents to handle the influence of different neighbors. SCN expands the agents' communication field with the chain propagation characteristics of graph neural networks (GNN). The main contributions and novelties are summarized as follows:

- A novel method, called MA-HCDP, is proposed to promote cooperation behaviors in environments with many agents.
- A hierarchical group network based on prior knowledge is designed to extract high-dimensional group-level state representation.
- A cognition difference network based on a variational autoencoder is designed to allow each agent to choose its neighbors adaptively to communicate.
- The effectiveness of MA-HCDP is evaluated in different tasks including cooperative navigation and group containment. Compared with existing methods, MA-HCDP shows a significant improvement in all the tasks, especially for the tasks with numerous agents.

The rest of this paper is organized as follows. In Section 2, we present the related works. In Section 3, we describe the background. In Section 4, we give the design procedure of the proposed method MA-HCDP, including HGN, CDN, and SCN. In Section 5, representative simulations are carried out in several scenarios. In Section 6, the discussion for the simulation results is presented. Finally, conclusions are summarized in Section 7.

## 2. Related Works

The multiagent deep deterministic policy gradient (MADDPG) [9] is extended from the deep deterministic policy gradient (DDPG) [7] to multiagent systems for mixed cooperative–competitive environments. A counterfactual multiagent (COMA) [10] computes a counterfactual advantage function to handle the problem of multiagent credit assignment. They adopt a common paradigm of centralized learning with decentralized execution (CTDE) to enhance cooperative behaviors of agents. Although MADDPG and COMA can improve the agents' cooperation ability, they do not consider the complex interactions among the agents. They aggregate observations of all the agents and never distinguish different groups of agents, thus limiting the cooperation of agents.

To address the limitation, the agent grouping method [12] employs a two-level graph neural network to model the interagent and intergroup relationships effectively. However, it ignores the communication relationship between the agents in the same group. The authors of [13] designed a two-level attention network to distinguish the different semantic meanings of observation. Nonetheless, it does not consider the communication contents.

To deal with the problem of communication, a feedforward deep neural network is adopted in [14] to map all agents' inputs to their actions. Each agent can have access to an implicit communication channel to receive other agents' states. A bidirectionally

coordinated network (BiCNet) [19] based on the actor–critic model adopts bidirectional recurrent networks to achieve mutual communication between agents. Master–slave [20] is a communication architecture for real-time strategy games, where the action of each slave agent consists of information from the slave agents and master agent. Although these methods can promote cooperation behaviors of agents, they need the global states of the environment during the training, which is not applicable in partially observable environments. Furthermore, these methods take all the communication among the agents into consideration, but they ignore the influence of redundant communication caused by too many communication objects.

To address this issue, some methods based on the attention mechanism appeared [15–18]. Nonetheless, each agent within the communication range will be assigned an attention coefficient with attention mechanism. The total amount of communication is not decreased since irrelevant agents can still obtain the attention coefficients and be included in the total communication amount. The soft attention mechanism usually assigns small but nonzero attention coefficients to irrelevant agents, which weakens the attention assigned to the significant agents.

Based on the above analyses, MA-HCDP is proposed to handle state representations of different group of agents and filter irrelevant agents to promote agents' cooperation behavior. To the best of our knowledge, none of existing work in MARL tackles simultaneously the problem of agents grouping, redundant communication information processing, and agent interaction processing like our MA-HCDP.

## 3. Background

In this section, the definition of partially observable Markov games (POMG) is presented firstly to describe the decision process of agents in partially observable environments. Then, the preliminary of reinforcement learning (RL) methods applied in MA-HCDP is introduced, including basic concepts and the actor–critic framework of RL and PPO. Moreover, the attention mechanism for handling the interaction among the agents is presented. Finally, the basic idea of variational autoencoding is introduced.

### 3.1. Partially Observable Markov Games

In this paper, agents need to cooperate with each other to complete different tasks in partially observable environments, which are considered as partially observable Markov games that are an extension of Markov games [23]. They are defined by environment state $S^t$, action spaces $A^t = [a_1^t, \cdots, a_N^t]$ where $N$ is the number of agents, $a_i^t$ is the action of agent $i$ at time $t$, and observation spaces $O^t = [o_1^t, \ldots, o_N^t]$. Each agent $i$ learns a policy $\pi^i : o_i^t \rightarrow P_a(a_i^t)$, which maps each agent's observation to a distribution over its set of actions. Then, the next states are produced according to the transition function $T : S^t \times a_1^t \times \cdots \times a_N^t \rightarrow P_t(S')$. Each agent $i$ obtains rewards $r_i$ as a function of the state space and action spaces: $S^t \times a_1^t \times \ldots \times a_N^t \rightarrow \mathbb{R}$. The goal of each agent is to maximize following expected discounted returns with the policy $\pi^i$:

$$J_i(\pi^i) = E_{a_1 \sim \pi^1, \ldots, a_N \sim \pi^N, s \sim T} \left[ \sum_{t=0}^{\infty} \gamma^t r_i^t (S^t, a_1^t, \ldots, a_N^t) \right] \tag{1}$$

where $E[\cdot]$ represents the expectation. $r_i^t$ represents the reward that agent $i$ obtains at time $t$ and $\gamma \in [0, 1]$ denotes the discount factor determining the importance of future rewards. If $\gamma = 0$, the agent will be completely myopic and only learn about actions that produce an immediate reward. If $\gamma = 1$, the agent will evaluate each of its actions based on the sum total of all of its future rewards.

### 3.2. Reinforcement Learning

Reinforcement learning [24] is adopted to solve special POMG problems where $N = 1$. It is a machine learning approach to solve sequential decision-making problems. Policy gradient methods are the popular choice for a variety of RL tasks. The policy $\pi$ realized by

a neural network with parameters $\theta$ is denoted as a policy $\pi_\theta$. Its objective is to directly adjust the parameters $\theta$ of the policy in order to maximize the objective $J_i(\pi_\theta^i)$ by taking steps in the direction of $\nabla_\theta J_i(\pi_\theta^i)$:

$$\nabla_\theta J_i(\pi_\theta^i) = \nabla_\theta \log\left(\pi_\theta^i(a_t|S_t)\right) \sum_{t'=t}^{\infty} \gamma^{t'-t} r_i^{t'}\left(s_i^{t'}, a_i^{t'}\right) \tag{2}$$

The actor–critic framework is one of the most effective RL frameworks. The key feature of the framework lies in two functions: policy function and value function. The policy function is known as the actor function, because it is used to select actions. The value function is known as the critic, because it criticizes the actions made by the actor. They reinforce each other. Specifically, the actor selects actions, then the actions is evaluated by the critic. Then, the critic updates the actor toward the right direction. This mutual reinforcement behavior enables policies to converge faster.

The proximal policy optimization algorithm (PPO) [25] is a novel policy gradient method. Considering the advantage of the actor–critic framework, we adopt PPO based on the actor–critic framework as the basic training algorithm in this paper. The objective function of PPO changes from (2) to (4) for a single-agent environment:

$$l_t(\theta) = \frac{\pi_\theta(a_t|s_t)}{\pi_{\theta k}(a_t|s_t)} \tag{3}$$

$l_t(\theta)$ denotes the likelihood ratio. $\pi_{\theta k}$ denotes the policy of the agent before $k$ steps. Then, the objective function is optimized according to the following equations:

$$L_\pi(\theta) = E[\min(l_t(\theta)\hat{A}_t^{\theta k}(s_t, a_t),\ clip(l_t(\theta), 1-\varepsilon, 1+\varepsilon)\hat{A}_t^{\theta k}(s_t, a_t)] \tag{4}$$

where $\hat{A}_t^{\theta k}(s_t, a_t)$ is the generalized advantage estimate (GAE) and $clip(l_t(\theta), 1-\varepsilon, 1+\varepsilon)$ clips $l_t(\theta)$ in the interval $[1-\varepsilon, 1+\varepsilon]$.

To train agents to learn cooperative behaviors, PPO is extended to multiagent environments in MA-HCDP, and the details are introduced in Section 4.4.

### 3.3. Attention Mechanism

The attention mechanism has been adopted in many fields [26–28]. For the attention mechanism, its inputs are composed of several input vectors $[B_1, B_2, \cdots, B_i, \cdots, B_K]$ and a target vector $Y^V$. The attention weight is related to a user-defined function $f(Y^V, B_i)$. $P$ is a weighted sum of each vector $B_i$ according to the normalized attention weight $w^i$:

$$\begin{aligned} w^i &= \frac{\exp(f(Y^V, B_i))}{\sum_{i=1}^{K} \exp(f(Y^V, B_i))} \\ P &= \sum_{i=1}^{K} w^i B_i \end{aligned} \tag{5}$$

Note that since

$$\sum_{i=1}^{K} w^i = 1,$$

the attention weight vector $W \stackrel{\Delta}{=} [w^1, w^2, \cdots, w^i, \cdots, w^K]$ denotes a probability distribution.

In this paper, the attention mechanism is enhanced to handle the relationship among the agents. The input vectors are the hidden states of the neighbor agents. The target vectors are the states of the center agent. The user-defined function is a feedforward neural network. The details are presented in Section 4.

### 3.4. Variational Autoencoder

The variational autoencoder [21] is a probabilistic latent variable model that relates an observed variable vector $x$ to a latent variable vector $z$ by a posterior distribution $q(z|x)$. In this paper, the posterior distribution $q(z|x)$ represents the understanding of the

environment for agents. It can be adopted as a basis to judge whether communication is needed between agents. The details are presented in Section 4.

## 4. Multiagent Hierarchical Cognition Difference Policy

In this section, a multiagent hierarchical cognition difference policy (MA-HCDP) is proposed as shown in Figure 1, including a hierarchical group network (HGN), a cognition difference network (CDN), and a soft communication network (SCN). First, HGN uses prior knowledge or data to cluster all the agents into different groups (including a friendly group, enemy group, and object group) and adopts attention mechanism [27] to extract different high-dimensional state representations of different groups. Then, CDN is responsible for filtering irrelevant agents to reduce redundant information. Next, the filtered communication status among the agents is modeled as a graph $G$ according to the CDN results. SCN is responsible for the weight distribution of the filtered agents to handle different neighbors' influence. Finally, the captured states of different groups are subsequently used to update the critic and actor network.

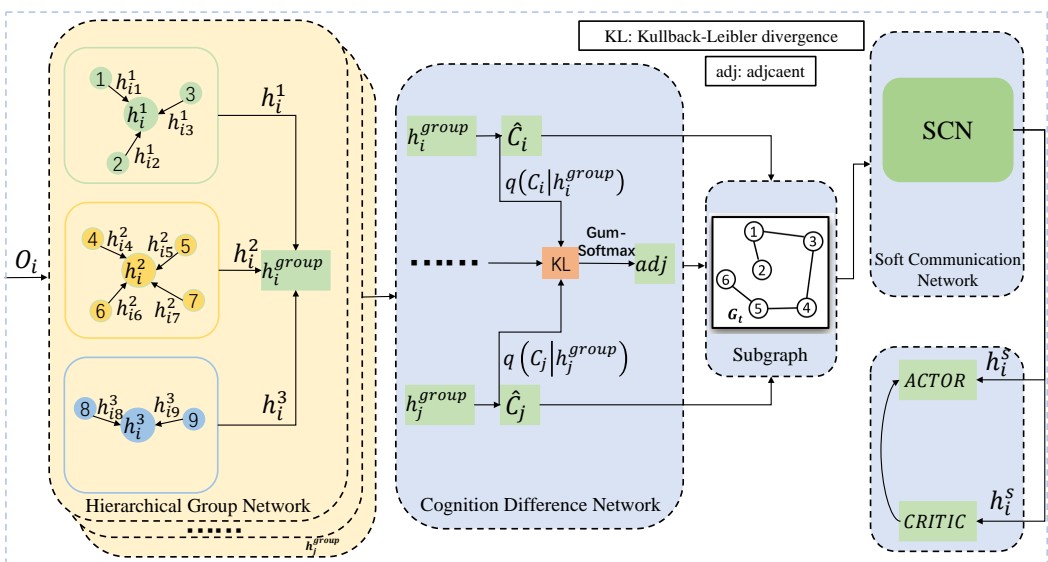

**Figure 1.** Architecture of MA-HCDP. Suppose there are nine entities around agent *i*. HGN divides the entities into different groups and adopts attention mechanism to extract high-dimensional state representations. Then, CDN is responsible for filtering irrelevant agents to reduce redundant information. Next, the remaining agents are modeled as a graph *G*. SCN is responsible for the weight distribution of the remaining agents. Finally, the captured states of different groups are subsequently used to update critic and actor networks.

### 4.1. Hierarchical Group Network

In this part, the influence of neighbor agents is analyzed firstly. Then, details of the hierarchical group network are presented.

In partially observable environments with many agents, there are many neighbors within the agents' observation range. In general, these neighbors can be divided into different groups based on prior knowledge, including a friend group, enemy group, and object group (obstacles or common goal). For instance, agents can be categorized into two groups, the friend group (agents) and the object group (obstacles or common goals), in a cooperative task. Different state representations of different groups have different influences on agents' policies. If an agent does not distinguish between states of the different groups, the influence of the states of different groups on the agent will be stacked together, thereby affecting its policy. Therefore, a hierarchical group network (HGN) is proposed to handle the different influences of state representations of different groups on the agent, which is shown in Figure 1.

In this paper, the other entities such as agents or obstacles within observation range of each agent $i$ ($i = 1, \ldots, N$) are categorized into $H$ groups, and agent $i$ is set as the center of each group. Note that agent $i$ has a local observation $o_i^t = \left[o_i^{t1}, \ldots, o_i^{th}, \ldots, o_i^{tH}\right]$ where $o_i^{th} = \left[o_{i0}^{th}, o_{i1}^{th}, \ldots, o_{ij}^{th}, \ldots, o_{iN^h}^{th}\right]$, $N^h$ is the number of entities in group $h$, and $h = 1, \ldots, H$. $o_{i0}^{th}$ is the own state of agent $i$.

Then, the attention mechanism is adopted by agent $i$ to obtain different state representations $h_i^h$ of different groups. It can enable agents to handle the states of the other entities in a group effectively. As shown in Figure 2, by using linear weight matrices $W^Q$, $W^K$, $W^V$, the states $o_{ij}^{th}$ of entity $j$ in group $h$ for agent $i$ are transformed to a different space including *query* $Q_j = W^Q o_{ij}^{th}$, *key* $K_j = W^K o_{ij}^{th}$ and *value* $V_j = W^V o_{ij}^{th}$. After receiving the (*query*, *value*) pair from the entities in group $h$, the attention coefficient $a_{ij}^h$ from entity $j$ in group $h$ to agent $i$ is computed. Then, the state representations of entities in group $h$ for agent $i$ are aggregated according to attention coefficients $a_{ij}^h$:

$$e_{ij}^h = \frac{\left(W^Q o_{ij}^{th}\right)\left(W^K o_{i0}^{th}\right)}{d_K}, \ a_{ij}^h = \frac{\exp\left(e_{ij}^h\right)}{\sum_{j \in N^h} \exp\left(e_{ij}^h\right)} \tag{6}$$

$$h_i^h = \sigma\left(\sum_{j \in N^h} a_{ij}^h V_j\right) \tag{7}$$

where $d_K$ is the dimensionality of *key* and $h_i^h$ is the aggregated states from other entities in group $h$ with nonlinear activation function $\sigma$.

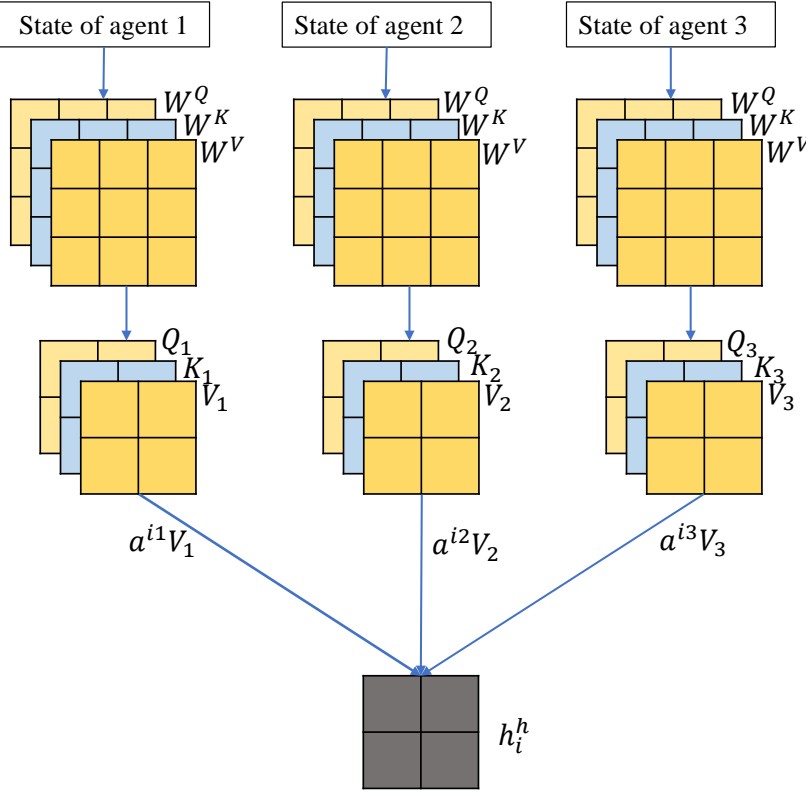

**Figure 2.** Architecture of one group attention. Suppose there are three agents around agent $i$ in a group. State representations of agents in group $h$ for agent $i$ are aggregated according to attention coefficients $a^{ij}$.

After state representation of all groups are obtained, they are aggregated to form a high-dimensional group-level state representation for agents' policy learning.

$$h_i^{group} = \sum_{h=1}^{H} h_i^h \tag{8}$$

With HGN, the entities are divided into different groups, which enables the agents to handle the state of different groups separately. The attention mechanism in HGN allows the agents to process the different influence of the other entities in a group, which helps the agents to understand the environment more effectively.

*4.2. Cognition Difference Network*

In this part, we first analyze the importance of redundant agents. Then, the design process of the cognition difference network is presented.

In environments with a large number of agents, if an agent communicates with all of its neighbors of the groups, redundant information will affect the agent's decision. When the number of agents in the environment increases, the communication among agents become more complicated, which makes the influence of redundant information more serious. Therefore, it is necessary for the agents to process redundant information.

Artificial rules [11] or a soft attention mechanism [15–18] are often used to process communication among the agents. The methods with artificial rules require strong prior knowledge of the environment and may not be suitable for application in complex environments. The methods with a soft attention mechanism assign attention coefficients to all agents within the communication range. The total amount of communication is not decreased since irrelevant agents can still obtain the attention coefficient and be included in the total communication amount. The soft attention mechanism usually assigns small but nonzero attention coefficients to irrelevant agents, which essentially weakens the attention assigned to the significant agents. Therefore, a cognition difference network (CDN) is designed to filter redundant agents to reduce redundant information.

First, we construct the communication status between agents as a graph, where each node represents a single agent, and all nodes are connected in pairs by default. $G^t = (V, E^t)$ represents the communication status between the agents. In particular, $V = \{1, \ldots, N\}$ is a set of the agents. $E^t \subseteq V \times V$ denotes the edge set at time $t$. $h^{t,group}$ is a set of group state representations at time $t$, $h^{t,group} = \left\{ \vec{h}_1^{t,group}, \vec{h}_2^{t,group}, ..., \vec{h}_N^{t,group} \right\}, \overrightarrow{h_i^{t,group}} \in \mathbb{R}^L$, where $L$ represents the state representation dimension of each node. Moreover, $N_i^t$ represents a set of neighbors of node $i$ at time $t$ in the graph. Agent $j \in N_i^t$ if $\left\| p_i^t - p_j^t \right\|_2 < r_c$ where $r_c$ is communication range and $\|\cdot\|_2$ is a two-dimensional Euclidean norm. As indicated by (9), there is a time-varying adjacency matrix $A^t$ where $a_{ij}^t = 1$ if $j \in N_i^t$ otherwise $a_{ij}^t = 0$.

$$a_{ij}^t = \begin{cases} 1 & if \ dist\left(p_i^t, p_j^t\right) \leq r_c \ or \ i = j \\ 0 & if \ dist\left(p_i^t, p_j^t\right) > r_c \end{cases} \tag{9}$$

where *dist* denotes a two-dimensional Euclidean norm that can calculate the distance between two agents.

Then, we introduce definitions of cognitive difference based on an assumption below to filter neighbor agents more effectively.

**Definition 1.** *The cognition of an agent is its understanding of the local environment. It contains the states of all entities within its observation range, or the high-level hidden states or distributions extracted from these states (e.g., learned with deep neural networks (DNNs)).*

**Definition 2.** *The cognition difference is the difference of high-level hidden states or distributions between the agents measured by n-dimensional norm or distribution measures.*

**Assumption 1.** *The cognition of each agent can be represented by a vector $C=[c_1, \ldots, c_k]$ or a distribution $p\left(C_i \middle| h_i^{group}\right)$ obeying Gaussian distribution.*

Under the above definitions and assumption, if the cognition difference between agent $i$ and agent $j$ is relatively large, agents' perceptions of the environment are quite different. In other words, the state of agent $j$ is noise or disturbance to agent $i$, which will affect the policy of agent $i$, so agent $i$ does not need to communicate with agent $j$.

After analyzing the role of the cognition difference, we need to solve two problems, including the representation of the cognition and the measurement of the cognition difference.

The representation of the cognition of agent $i$ is denoted as cognition vector $C_i$; it is based on the states of all entities within its observation range. The common methods are to adopt directly multilayer perception (MLP) [9–11] or a graph convolution network (GCN) [15,29,30] to extract features of the observations as the cognition vector $C_i$. However, the cognitive vectors extracted by these methods are essentially the result of single-valued mapping from vector to vector. These methods cannot fully decouple the observations' factors, such as position or velocity. Therefore, the cognition vector extracted by MLP or GCN is not appropriate for the cognition of the environment.

In order to effectively represent the cognitive vector, we adopt a probability distribution method. Specifically, the cognition vector $C_i$ is sampled with posterior distribution from group states $h_i^{group}$ and $C_i$ is inferred with:

$$p\left(C_i \middle| h_i^{group}\right) = \frac{p\left(h_i^{group}|C_i\right)p(C_i)}{p\left(h_i^{group}\right)} = \frac{p\left(h_i^{group}|C_i\right)p(C_i)}{\int p\left(h_i^{group}|C_i\right)p(C_i)dC_i} \tag{10}$$

where $p\left(h_i^{group}|C_i\right)$ is a reconstruction process from the cognition vector $C_i$. $p\left(C_i \middle| h_i^{group}\right)$ is the cogniton representation for agent $i$.

However, the above equation is diffcult to calculate directly. Hence, we approximate $p\left(C_i \middle| h_i^{group}\right)$ by another tractable distribution $q\left(C_i \middle| h_i^{group}\right)$. The restriction is that $q\left(C_i \middle| h_i^{group}\right)$ needs to be close to $p\left(C_i \middle| h_i^{group}\right)$. We achieve it by minimizing the following KL divergence:

$$\min KL\left(q\left(C_i \middle| h_i^{group}\right) \middle\| p\left(C_i \middle| h_i^{group}\right)\right) \tag{11}$$

which is equal to the maximum of the evidence lower bound (ELBO) [31]:

$$\max E_{q\left(C_i \middle| h_i^{group}\right)} \log p\left(C_i \middle| h_i^{group}\right) - KL\left(q\left(C_i \middle| h_i^{group}\right) \middle\| p(C_i)\right) \tag{12}$$

Note that the former is the reconstruction likelihood, and the latter is used to ensure that $q\left(C_i \middle| h_i^{group}\right)$ and the true prior distribution $p(C_i)$ are as similar as possible. This process can be modeled as a VAE [21] as shown in Figure 3. Specifically, a mapping $q\left(\widehat{C}_i \middle| h_i^{group}; \chi\right)$ from $h_i^{group}$ to $\widehat{C}_i$ is obtained in the encoder of VAE where $\chi$ are the parameters of DNNs. Next, we adopt the "reparameterization trick" to sample $\varepsilon$ from a unit Gaussian. Then, $\widehat{C}_i$ with mean $\mu_{C_i}$ and variance $\sigma_{C_i}$ is generated: $\widehat{C}_i = \mu_{C_i} + \sigma_{C_i} \odot \varepsilon$ where $\varepsilon \sim N(0,1)$. A distribution mapping $p\left(\widehat{h_i^{group}} \middle| \widehat{C}_i; \chi\right)$ from $\widehat{C}_i$ back to $\widehat{h_i^{group}}$ is learned in the decoder of the VAE. Based on the above analyses, the loss function for training the VAE is:

$$\min L2\left(h_i^{group}, \widehat{h_i^{group}}; \chi\right) + KL\left(q\left(\widehat{C}_i \middle| h_i^{group}; \chi\right) \middle\| p(C)\right) \tag{13}$$

where $\widehat{C}_i$ is the result of the cognition representation of agent $i$ and the learned distribution $q\left(\widehat{C}_i \middle| h_i^{group}; \chi\right)$ is an approximation of $p\left(C_i \middle| h_i^{group}\right)$.

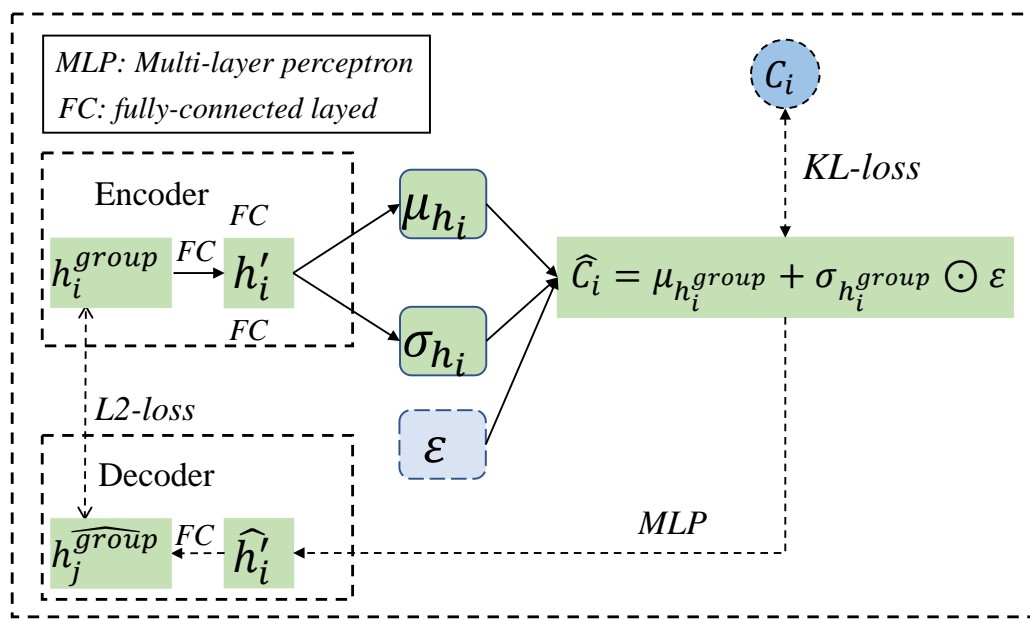

**Figure 3.** Architecture of variational autoencode. The agent understanding for the environment is obtained via variational autoencoder (VAE). L2-loss is the reconstruction loss function and KL-Loss is the loss function for VAE training.

For the measurement of difference, since $\widehat{C}_i$ is sampled from the learned distribution, the distribution is more general in the environment's cognition than $\widehat{C}_i$. Therefore, we choose the difference between the distributions as the cognition difference. To calculate the cognition difference between agent $i$ and agent $j$, KL divergence is adopted to calculate a score $C_d^{ij}$ that measures the difference.

$$C_d^{ij} = KL\left(q\left(\widehat{C}_i\middle|h_i^{group};\chi\right)\middle\|q\left(\widehat{C}_j\middle|h_j^{group};\chi\right)\right) \tag{14}$$

We need to output a one-hot vector based on the cognition difference to determine whether the edge between agent $i$ and $j$ exist in the graph $G^t$ or which agents need to be communicated. However, the backpropagation of gradients cannot be achieved in the process of outputting the one-hot vector due to the sampling process. Therefore, the Gumbel-Softmax function [32] is adopted to solve it:

$$W_c^{i,j} = gum(f\left(KL\left(q\left(\widehat{C}_i\middle|h_i^{group};\chi\right)\middle\|q\left(\widehat{C}_j\middle|h_j^{group};\chi\right)\right)\right)) \tag{15}$$

where $gum(\cdot)$ represents the Gumbel-Softmax function.

With the cognition difference and the sample process of Gumbel-Softmax, we can get a subgraph $G_i^{'}$ for agent $i$, where agent $i$ only connects with agents that need to communicate.

With CDN, the agent understanding for the environment is obtained via VAE. Then, agents that have different understanding for the environment are defined as irrelevant agents and filtered out, which can reduce redundant information to enable agents to cooperate better.

*4.3. Soft Communication Network*

In this part, the influence of filtered neighbor agents is analyzed. Then, details of the soft communication network are presented. Furthermore, the process of enlarging the communication field is described.

Actually, in addition to processing redundant agents, the filtered neighbor agents of agent $i$ need to be treated differently to promote cooperation. Different neighbors have different cognition of the environment, and this will have different influences on agent $i$. For instance, to an agent, agents closer to it may have more influence than agents further

away, which means the agent should take into the influence of different agents. Therefore, the graph attention mechanism [33] is adopted in SCN to enable agents to handle the filtered neighbor agents' states differently.

Specifically, SCN operates on graph-structured data and obtains the features of each graph node by aggregating the states of its neighbors. As shown in Figure 4, the attention coefficient $e_{ij}$ from agent $j$ to agent $i$ and its normalized form $\alpha_{ij}$ are calculated with the hidden states of the agents:

$$e_{ij} = a_G^k\left(W_G^k\widehat{C}_i, W_G^k\widehat{C}_j\right) \tag{16}$$

$$\alpha_{ij} = \text{softmax}(e_{ij}) = \frac{\exp\left(\text{LeakyReLU}\left(e_{ij}\right)\right)}{\sum_{k\in N_i}\exp\left(\text{LeakyReLU}\left(e_{ij}\right)\right)} \tag{17}$$

where $W_G^k$ is a linear learnable matrix, $a_G^k$ is a single-layer feedforward neural network, and *LeakyReLU* is a nonlinear activation function. [33] shows that the import of multihead attention is helpful to stabilize the learning process of the attention coefficients. Therefore, the aggregation states of agent $i$ with multihead attention at $t$ is given by:

$$h_i^s = \bigg\|_{m=1}^M \sigma\left(\sum_{j\in N_i}\alpha_{ij}^m W^m\widehat{C}_j\right) \tag{18}$$

where $N_i$ denotes the filtered neighbor agents for agent $i$. $\|$ is the concatenation operator and $M$ denotes the number of the heads. $W^m$ is the weight matrix of the $m$th linear transformation and $\alpha_{ij}^m$ is the normalized attention coefficient of the $m$th attention head.

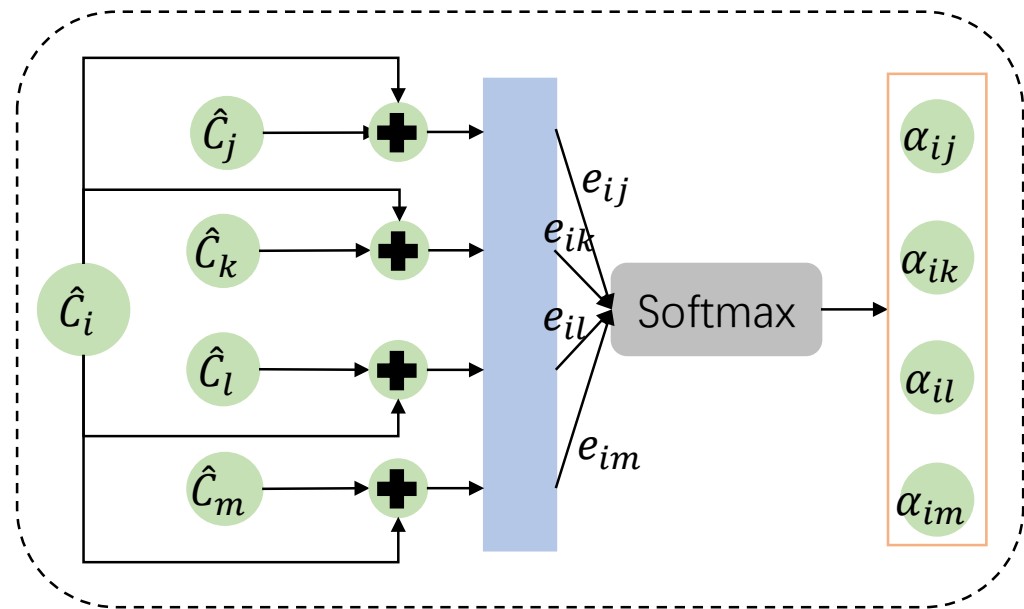

**Figure 4.** Architecture of attention. Different states of neighbors of agent $i$ ($\widehat{C}_j, \widehat{C}_k, \widehat{C}_l, \widehat{C}_m$) are concentrated with agent $i$; then, the attention coefficient $e_{ij}$ is calculated. The normalized form of $e_{ij}$ is $\alpha_{ij}$, and it is used as the weight of state concentration.

Moreover, the chain propagation characteristic of the graph convolution is adopted in SCN to enlarge the communication field. As described in Figure 5, agent 4 can get the states from its neighbors (agent 3 and agent 5) with one SCN layer. With two layers, agent 4 further obtains the states of its neighbors' neighbors (agent 1 and agent 6). By stacking a third layer, agent 4 can finally obtain all the agents' states. Therefore, multiple SCN layers can be utilized to enlarge the communication field of agent 4. Although multiple stacked SCN layers can take more additional structure information from the neighbor nodes of

the center node into consideration, the cost of computing is greatly increased. Therefore, the number of SCN layers is set to three in this paper.

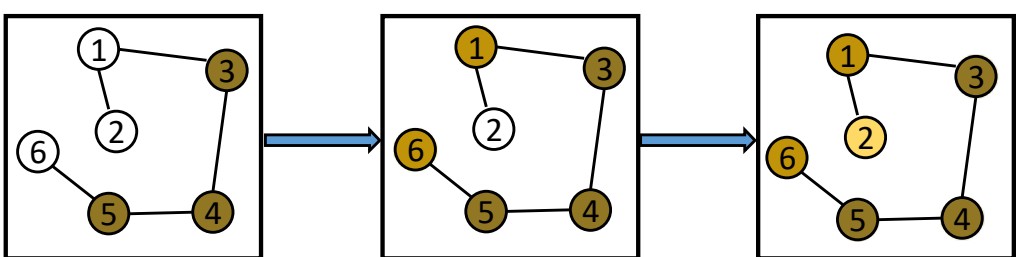

**Figure 5.** Enlarging respective fields. With stacked layers, agent 4 first obtains the information of agents 3 and agent 5, then obtains the information of agents 1 and 6, and finally obtains the information of agent 2.

With SCN, each agent can handle different cognition of different neighbor agents. The communication field is enlarged with the chain propagation characteristic of graph, which promotes cooperation behaviors of the agents.

### 4.4. Training Method

After extraction by SCN, $h_i^s$ is utilized to optimize the policy of agents. PPO is adopted to train the agents based on an actor–critic framework. With HGN, the agent can obtain high-dimensional state representation of different groups. Owing to CDN and SCN, each agent can choose the agents to communicate with and assign different weights to them. The objective function of PPO as shown in (4) is changed as (21) after $h_i^s$ extracted from CDN and SCN. Then, PPO is trained by minimizing a total loss $L_{total}$, which is conducted by the weighted summation of value loss $L_V(w)$, action loss $L_\pi(\theta)$, action entropy $H(\theta)$, and cognition difference loss $L_{cd}(\chi)$:

$$L_V(w) = E\left[\left(\begin{array}{c} Q(O_1, O_2, \cdots, O_N, a) \\ -Q(O_1, O_2, \cdots, O_N, a; w) \end{array}\right)^2\right] \tag{19}$$

$$l_t(\theta) = \frac{\pi_\theta(a_t | (O_1, O_2, \cdots, O_N))}{\pi_{\theta^k}(a_t | (O_1, O_2, \cdots, O_N))} \tag{20}$$

$$L_\pi(\theta) = E[\min(l_t(\theta)\hat{A}_t^{\theta^k}(O_1, O_2, \cdots, O_N), \ clip(l_t(\theta), 1-\varepsilon, 1+\varepsilon)\hat{A}_t^{\theta^k}(O_1, O_2, \cdots, O_N)] \tag{21}$$

$$H(\theta) = -\sum \pi_\theta(a_t | O_1, O_2, \cdots, O_N) \log(\pi_\theta(a_t | O_1, O_2, \cdots, O_N)) \tag{22}$$

$$L_{cd}(\chi) = E\left[\left(L2\left(h_i^{group}, \widehat{h_i^{group}}; \chi\right) + KL\left(q\left(\hat{C}_i \Big| h_i^{group}; \chi\right) \| p(C)\right)\right)\right] \tag{23}$$

$$L_{total} = \beta_1 L_V(w) + \beta_2 L_\pi(\theta) - \beta_3 H(\theta) + \beta_4 L_{cd}(\chi) \tag{24}$$

where $\beta_i$ is the weight coefficient of the loss function. The action entropy $H(\theta)$ is specially designed to encourage exploration for agents by penalizing the entropy of actor $\pi_\theta(a_t | O_1, O_2, \cdots, O_N)$. The implementation details are presented in Algorithm 1.

---

**Algorithm 1** MA-HCDP.

---

**Input**: agent's state $o_i$

**Initialization**: Initialize actor $\theta_a$, critic $\theta_c$, and old actor $\theta_a^{old}$ network

1: **for** Episode 1 **to** M **do**

2:     Run policy $\pi_{\theta^a}(o_1, \ldots, o_N)$ for $T$ time-steps for each agent, collecting $\{o, a, r\}$ where
       $o = (o_1, \cdots, o_N)$, $a = (a_1, \cdots a_N)$ and $r = (r_1, \cdots, r_N)$
       Estimate advantages function $A_t$
       $\pi_{\theta_a^{old}} \leftarrow \pi_{\theta^a}$

3:     **for** k = 1 **to** 4 **do**

4:         Calculate value loss: $L_v(w)$
           $$L_v(w) = \left(Q - Q_{final}^t(w)\right)^2$$
           Calculate action loss: $L_\pi(\theta)$
           $$L_\pi(\theta) = E[\min(l_t(\theta)\hat{A}_t^w(x, a_1, \ldots, a_N),$$
           $$clip(l_t(\theta), 1 - \varepsilon, 1 + \varepsilon)\hat{A}_t^w(x_{t,1}, \ldots, a_N)]$$
           Calculate entropy loss: $H(\theta)$
           $$H(\theta) = -\sum \pi_\theta \log(\pi_\theta)$$
           Calculate cognition difference loss: $L_{cd}(\chi)$
           $$L_{cd}(\chi) = E\left[\left(L2\left(h_i^{group}, \widehat{h_i^{group}}; \chi\right) + KL\left(q\left(\hat{C}_i \Big| h_i^{group}; \chi\right) \| p(C)\right)\right)\right]$$
           Calculate total loss: $L_{total}$
           $$L_{total} = \beta_1 L_V(w) + \beta_2 L_\pi(\theta) - \beta_3 H(\theta) + \beta_4 L_{cd}(\chi)$$
           Update actor and critic network by minimizing $L_{total}$

5:     **end for**

6: **end for**

---

## 5. Simulation Results and Analysis

### 5.1. Simulation Settings

In this section, the performance of MA-HCDP is evaluated in two tasks, including cooperative navigation and group containment. The cooperative navigation task is designed to evaluate the effectiveness of MA-HCDP in handling states of differnt groups of agents. The group containment tasks is designed to verify the performance of MA-HCDP in reducing redundant communication information.

For all the tasks, the only way to obtain more information from the other agents is through limited communication. The map side length is 2 m, the detection range is 0.5 m, and the communication range is 1 m. The radius of an agent is 0.05 m and the radius of an obstacle is 0.1 m. The action space is discrete and each agent is able to control unit acceleration or deceleration in $X$ and $Y$ directions. The boundary condition for the environment is the four sides of the map. Each agent obtains $R_{cross}$ for crossing the boundary.

$$R_{cross} = \begin{cases} 0 & if\ x < 1.8 \\ 10(x - 1.8) & if\ 1.8 \leq x < 2 \\ \min\left(e^{2x-1.8}, 10\right) & if\ x \geq 2 \end{cases} \tag{25}$$

where $x$ is the abscissa or ordinate of the agent.

These tasks are implemented based on [9], where the agents can move around with a double integrator dynamics model. The details of the tasks will be presented in the following sections.

As baseline algorithms for comparing the performance, MADDPG [9] and TRANSFER [15] are taken into consideration to compare with our method MA-HCDP. MADDPG needs the state of all the agents during training to construct its critic network. It neither considers the influence of different groups nor the influence of redundant communication contents. TRANSFER adopts GCN and the soft attention mechanism to deal with the different influence of agents, but it ignores the influence of redundant communication contents.

A workstation is utilized for training and testing throughout the simulations, in which the processor is Intel(R) Xeon 8280L(2.6 GHz), the graphics card is Nvidia TITAN

RTX GPU(24 G), and the RAM size is 128 GB. These simulations are implemented in the multiagent particle environment (MAPE) [9], and these algorithms are trained with PyTorch 1.0. The parameters of the training process, the network, and MA-HCDP are given in Table 1. Specifically, the learning rate is a hyperparameter used in the training of neural networks. The max gradient normalization is used to clip the gradient to avoid exploding gradients. The episode, batch size, PPO epoch, the coefficient of value loss, policy loss, entropy, and VAE are hyperparameters used in the training of PPO. The number of attention heads in HGN is a hyperparameter used for the multihead attention mechanism of HGN.

**Table 1.** Parameters of training.

| Parameters | Value |
|---|---|
| Learning rate | 0.0001 |
| Max gradient normalization | 2 |
| Discount factor $\gamma$ | 0.99 |
| Coefficient of value loss function $\beta_1$ | 0.5 |
| Coefficient of policy loss function $\beta_1$ | 1 |
| Coefficient of entropy $\beta_3$ | 0.01 |
| Coefficient of VAE loss function $\beta_4$ | 0.01 |
| Episode | 20,000 |
| Batch size | 64 |
| PPO epoch | 4 |
| Number of attention heads in HGN | 3 |

*5.2. Cooperative Navigation*

For the cooperative navigation, there are $N$ agents and $N$ landmarks in the environment as shown in Figure 6a. The objective is for the agents to deploy themselves in a manner such that every agent reaches a distinct landmark. Note that we do not assign a particular landmark to each agent, but instead let the agents communicate with each other and develop a consensus as to who goes where. The compound reward function for this task is defined as follows:

$$
\begin{aligned}
R_{total}^i &= R_{dist}^i + R_{collision}^i + R_{cross} \\
R_{dist}^i &= -dist\left(p_i^t, p_i^g\right) \\
R_{collision}^i &= \begin{cases} 0 & if\ dist(p_i^t, p_j^t) > r_i + r_j \\ -1 & else \end{cases}
\end{aligned}
\tag{26}
$$

where $R_{dist}$ is the distance reward and $R_{collision}$ is the collision reward. $r_i$ represents the radius of agent $i$ and $p_i^g$ represents the goal position of agent $i$. The closer the agent is to the goal point, the greater $R_{dist}$ is.

In this paper, there are four different scenarios with different numbers of agents. Specifically, these scenarios include scenario (a) with 6 agents, scenario (b) with 15 agents, scenario (c) with 20 agents, and scenario (d) with 29 agents. MA-HCDP is compared with MADDPG and TRANSFER in these scenarios.

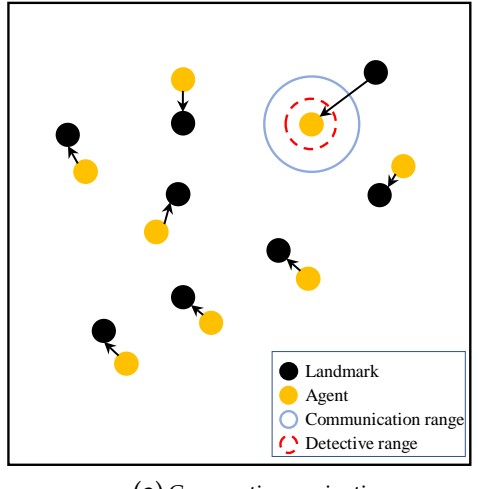
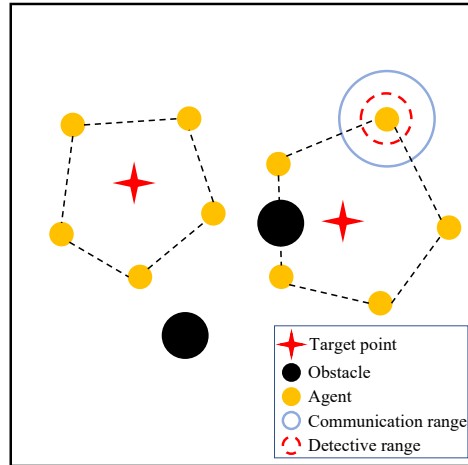

(**a**) Cooperative navigation

(**b**) Group containment

**Figure 6.** The illustration of simulation tasks.

The learning curves of all the approaches in terms of mean rewards are presented in Figure 7. The coordinate name of the $x$-axis indicates the time spent in training. The longer the time, the slower the method will converge. The coordinate name of the $x$-axis represents the reward obtained in an episode. The higher the reward, the better the performance of the method. As shown in Figure 7, MA-HCDP has a higher convergence rate and higher performance than MADDPG and TRANSFER. The results indicate that different influences of observations of different groups of agents can be handled by MA-HCDP. It demonstrates that the redundant communication information can be filtered effectively by MA-HCDP, which enable agents to learn more appropriate policy in fewer training episodes. Note that MADDPG converges faster than TRANSFER but converges to a minimum value. As a comparison, TRANSFER converges slower but obtain higher rewards than MADDPG. Although TRANSFER considers the influence of communication among different agents, it ignores the influence of redundant communication, which make agents trained with TRANSFER obtain higher rewards and converge slower than MADDPG. Due to the ignorance of the influence of redundant communication, although the agent trained by TRANSFER can learn appropriate policy, its performance and convergence speed are both weaker than MA-HCDP. The above analysis shows the effectiveness of our method in dealing with the influence of different groups of agents and redundant communication.

In addition to the training process data, the simulation results of 100 independent simulations for each scenario are presented in Table 2. The $t$-test value is adopted to evaluate the effectiveness of our method statistically. According to the mean value, standard deviation and the sample data for 100 tests, we calculate the $t$ value of MA-HCDP and the other methods on the same test scenario. "+" and "=" indicate that the index values obtained by the algorithm in this paper are superior and equal to the results of the other methods in the same test scenario in the two-tailed $t$-test with a significance level of %5.

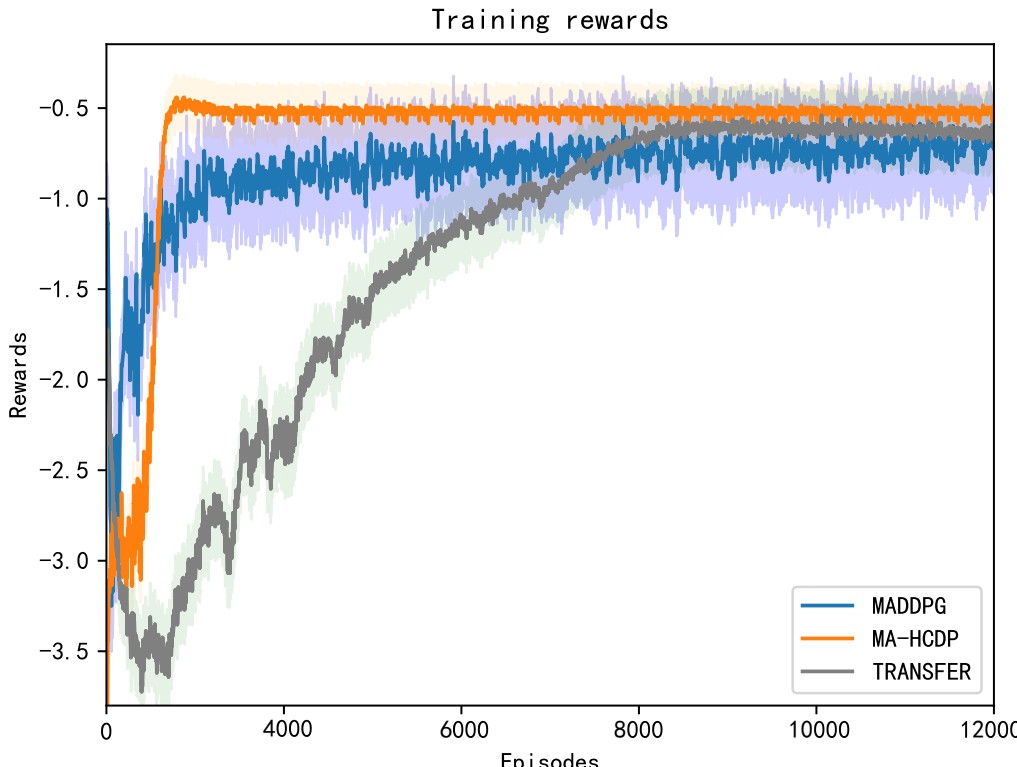

**Figure 7.** Simulation result in cooperative navigation with six agents.

As indicated by Table 2, our method obtains 18 optimal measurements during all the test scenarios. Note that the success rate of MADDPG in four scenarios is zero. This is because MADDPG does not consider the communication relationship among the agents and cannot handle partially observable environments. There is no significant performance difference between our method and TRANSFER in terms of success rate. The reason is that the task is simple, so TRANSFER and MA-HCDP can successfully complete the task. Despite this, our method is better than TRANSFER in terms of rewards and steps in scenarios with large number of agents. When the number of agents is equal to six, all methods except MADDPG present a similar performance. When the number of agents increases, our proposed MA-HCDP can obtain more rewards and take fewer steps than TRANSFER to complete the task. The reason is that agents need to communicate with more neighbor agents as the number of agents increases, and MA-HCDP can filter out redundant agents to promote multiagent cooperation behavior.

*5.3. Group Containment*

For the group containment, there are $n$ agents and $m$ landmarks in the environment as shown in Figure 6b. The relationship between $n$ and $m$ is defined as $\frac{n}{m} = k$, $k \in \mathbb{Z}$ where $\mathbb{Z}$ represents a set of positive integers. Based on the above restriction, two scenarios including 6 agents with 2 landmarks and 10 agents with 2 landmarks are designed to evaluate the performance of our scheme. In these scenarios, all the agents must be divided into two groups and distributed evenly around the two landmarks without collision. The compound reward function for this task is defined as follows:

$$
\begin{aligned}
R^i_{total} &= R^i_{dist} + R^i_{collision} + R_{cross} \\
R^i_{dist} &= -Hungarian\left(p^t_i, p^g_i\right) \\
R^i_{collision} &= \begin{cases} 0 & if\ dist(p^t_i, p^t_j) > r_i + r_j \\ -1 & else \end{cases}
\end{aligned}
\tag{27}
$$

where $p_i^t$ is the position of agent $i$ and $p_i^g$ is the position of goal. *Hungarian* represents the Hungarian algorithm, which is a combination optimization algorithm. The Hungarian algorithm is adopted to calculate the mean distance between the agents and the goal positions. With the reward setting, the distance reward for each agent is related to the other agents.

In this paper, there are two different scenarios with different number of agents for the group containment tasks. Specifically, these scenarios include scenario (a) with 6 agents and scenario (b) with 10 agents. MA-HCDP is compared with MADDPG and TRANSFER in these scenarios.

**Table 2.** Evaluation results of cooperative navigation.

| Method | | Scenario (a) with 6 Agents | | | Scenario (b) with 15 Agents | | |
|---|---|---|---|---|---|---|---|
| | | **Success Rate** | **Steps** | **Rewards** | **Success Rate** | **Steps** | **Rewards** |
| MADDPG | mean | 0 | 50 | −1.34 | 0 | 50 | −2.2 |
| | *t*-test | N/A(+) | −557.9873(+) | 94.2665(+) | N/A(+) | −522.7875(+) | 181.5395(+) |
| TRANSFER | mean | 100 | 14.2 | −0.52 | 97 | 17.18 | −0.61 |
| | *t*-test | N/A(=) | −1.2035(=) | 1.2131(=) | 1.0801(=) | −5.7893(+) | 6.3189(+) |
| MA-HCDP | mean | **100** | **14.01** | **−0.49** | **99** | **14.82** | **−0.53** |
| | *t*-test | N/A | N/A | N/A | N/A | N/A | N/A |
| Method | | Scenario (c) with 20 agents | | | Scenario (d) with 29 agents | | |
| | | Success rate | Steps | Rewards | Success rate | Steps | Rewards |
| MADDPG | mean | 0 | 50 | −3.48 | 0 | 50 | −3.91 |
| | *t*-test | N/A(+) | −472.6488(+) | 322.5309(+) | N/A(+) | −413.4042(+) | 373.1337(+) |
| TRANSFER | mean | 98 | 20.11 | −0.72 | 96 | 25.2 | −0.86 |
| | *t*-test | 0.9201(=) | −21.1968(+) | 9.2276(+) | 1.1241(=) | −38.0417(+) | 42.0352(+) |
| MA-HCDP | mean | **99** | **17.05** | **−0.64** | **98** | **20.06** | **−0.69** |
| | *t*-test | N/A | N/A | N/A | N/A | N/A | N/A |

The simulation results of MA-HCDP and TRANSFER are presented in Table 3. MA-HCDP obtain 12 optimal measurements during four scenarios and outperforms than TRANSFER and MADDPG in terms of steps and rewards. In this kind of competitive scenario, where there is a high demand for communication effectiveness, CDN in MA-HCDP can filter out unrelated agents, and SCN can process the information of filtered agents with attention weights to promote cooperation.

To further demonstrate the effectiveness of our method, we present the attention value distribution of agents in Figure 8. Take agent 2 and agent 5 in Figure 8a as an example. As shown in Figure 8, we can obtain the attention value distribution for agent 2 (Figure 8c) and agent 5 (Figure 8d). Note that agent 2 only communicates with agent 1 and agent 3, and agents 5 communicates with agent 6. The reason why agents 2 and 5 do not communicate is that agent 2 needs to go around landmark 1, while agent 5 needs to go around landmark 2. They have different perceptions of the environment and goals, so there is no need for communication. Based on the communication status, the attention coefficients are only assigned to those agents communicating with agent 2 and agent 5. Moreover, since different agents have different influences, they are assigned different attention coefficients. It demonstrates that MA-HCDP can filter out unrelated agents and process the information of different agents to promote cooperation.

**Table 3.** Evaluation results of group containment.

| Method | | Scenario (a) with 6 Agents | | | Scenario (b) with 10 Agents | | |
|---|---|---|---|---|---|---|---|
| | | Success Rate | Steps | Rewards | Success Rate | Steps | Rewards |
| MADDPG | mean | 0 | 80 | −1.68 | 0 | 80 | −4.12 |
| | *t*-test | N/A (+) | −583.7586 (+) | 252.3061 (+) | N/A (+) | −662.4380 (+) | 749.1802 (+) |
| TRANSFER | mean | 93 | 16.3 | −0.66 | 91 | 14.28 | −0.82 |
| | *t*-test | 87.3064 (+) | −38.9224 (+) | 32.3495 (+) | 103.8512 (+) | −33.4683 (+) | 36.2515 (+) |
| MA-HCDP | mean | **100** | **14.2** | **−0.56** | **100** | **12.9** | **−0.68** |
| | *t*-test | N/A | N/A | N/A | N/A | N/A | N/A |

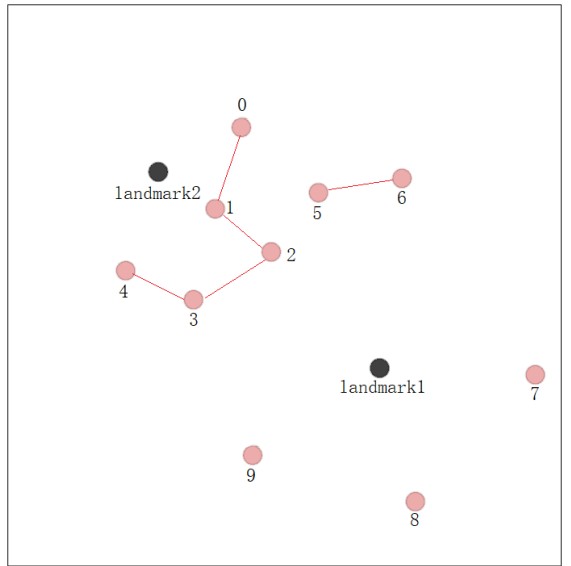

(**a**) The cooperation strategy in process.

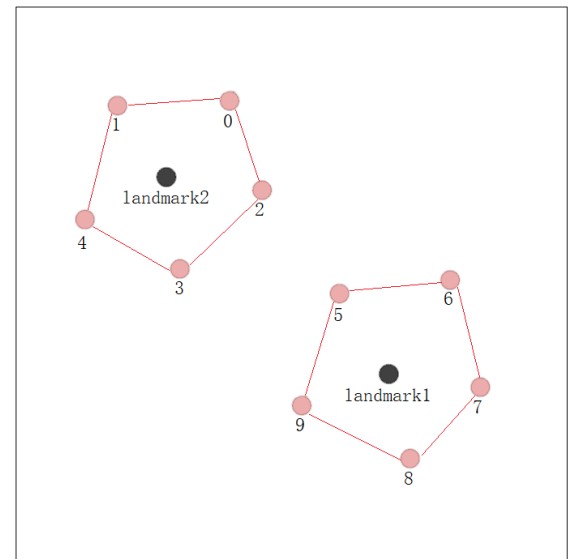

(**b**) The cooperation strategy at the end.

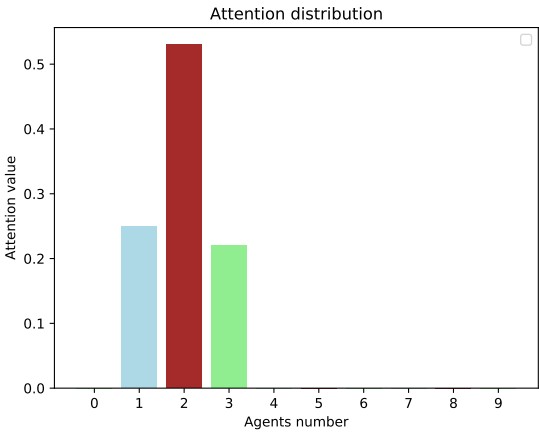

(**c**) Attention distribution for agent 2.

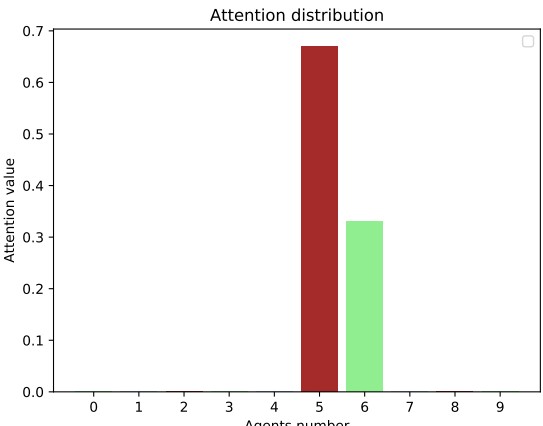

(**d**) Attention distribution for agent 5.

**Figure 8.** Attention distribution.

## 6. Discussion

In addition to the analyses above, there are some phenomena worth analyzing. Note that MADDPG converges faster than TRANSFER but converges to a minimum value. As shown in Table 2, the success rate of MADDPG is zero. This phenomenon may be caused by the limitation of MADDPG. Although MADDPG can obtain the state of all the agents during training, too much information makes the agents fall into a locally optimal situation and unable to learn appropriate policy. As a comparison, TRANSFER converges slower but obtains higher rewards than MADDPG. Although TRANSFER considers the influence of communication among different agents, it ignores the influence of redundant communication, which makes agents trained with TRANSFER obtain higher rewards and converge slower than MADDPG. Therefore, although the agents trained by TRANSFER can learn appropriate policy, its performance and convergence speed are both worse than our MA-HCDP.

As shown in Figure 8, agent 2 and agent 5 are both within the communication range of each other, but due to inconsistent cognition of the environment, there is no communication between them. Although agent 2 can obtain information from agent 0 and agent 4 and be influenced by them, the attention value from agent 0 and agent 4 to agent 2 is zero because the information and influence are obtained indirectly. Specifically, the adjacent matrix for agent 2 is:

$$
\begin{array}{ccccccccccc}
Index & 0 & 1 & 2 & 3 & 4 & 5 & 6 & 7 & 8 & 9 \\
2 & 0 & 1 & 1 & 1 & 0 & 1 & 0 & 0 & 0 & 0
\end{array}
$$

The state of agent 2 after stacked SCN layers are related to agent 0, agent 1, agent 3, and agent 4. Agent 2 communicates with agents outside the communication range indirectly through stacked SCN layers, rather than directly communicating with those agents. Therefore, the attention value from agent 0 and agent 4 to agent 2 is zero.

In addition to the above discussions, there are several limitations and possible validations for this study that may be addressed as future directions. HCN is responsible for dividing agents into different groups, but the grouping of the agents depends on prior knowledge. When the environment becomes too complex, it is a challenge to group the agents with prior knowledge. There is a certain difference between simulation environments and realistic environments. How to make methods that perform well in simulation environments also work well in realistic environment is a problem that we are working hard to study.

## 7. Conclusions

In this paper, we present a novel reinforcement learning method MA-HCDP for multiagent cooperation in environments with a large number of agents. Its key feature lays in a hierarchical group network (HGN), a cognition difference network (CDN), and a soft communication network (SCN). Specifically, HGN is responsible for dividing agents into different groups and extracting high-dimensional state representations of these groups. CDN is designed to extract the agents' understanding of the environment with VAE and filtering out irrelevant agents with KL divergence. SCN is responsible for handling different cognition of different neighbor agents with the attention mechanism and enlarging the communication field. Simulation results indicate that MA-HCDP can handle influences of different groups of agents and redundant communication and perform a satisfying strategy and adapt to environments with many agents. Our method has a considerable increase in terms of rewards and convergence compared with MADDPG and TRANSFER. The proposed method can handle the redundant information well. As shown in Figure 8, if the agents' information is redundant, there is no communicate between them, even if they are within the communication range of each other. Future work will take the adaptive grouping based on cognitive differences into consideration.

**Author Contributions:** Conceptualization, H.W.; methodology, H.W.; software, H.W.; validation, H.W.; formal analysis, H.W.; investigation, H.W.; resources, H.W.; data curation, H.W.; writing—original draft preparation, H.W.; writing—review and editing, H.W., Z.L., J.Y., and Z.P.; visualization, H.W.; supervision, Z.L.; project administration, Z.L.; funding acquisition, J.Y. All authors have read and agreed to the published version of the manuscript.

**Funding:** This research was funded by the National Key Research and Development Program of China under Grant 2018AAA0102402, in part by the external cooperation key project of the Chinese Academy of Sciences No. 173211KYSB20200002 and Innovation Academy for Light-duty Gas Turbine, Chinese Academy of Sciences, No.CXYJJ19-ZD-02 and No.CXYJJ20-QN-05.

**Conflicts of Interest:** The authors declare no conflict of interest.

## Abbreviations

The following abbreviations are used in this manuscript:

| | |
|---|---|
| MA-HCDP | Multiagent hierarchical cognition difference policy |
| HGN | Hierarchical group network |
| CDN | Cognition difference network |
| SCN | Soft communication network |
| VAE | Variational autoencoder |
| GNN | Graph neural networks |
| PPO | Proximal policy optimization |
| POMG | Partially observable Markov games |
| RL | Reinforcement learning |

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
