# Peer review of "Multiagent Hierarchical Cognition Difference Policy for Multiagent Cooperation"

_algorithms, doi:10.3390/a14030098_

Round 1

Reviewer 1 Report

The paper presents a novel reinforcement learning approach called MA-HCDP to be used in multi-agent cooperation in environments with a large number of agents. In this approach, the agents are grouped so that they can communicate in a more efficient way inside the group identifying the “most interesting neighbors” (those sharing the same goal and presenting a similar cognitive state, meaning that they are in a state similar to the one of the interested agent). The approach is then compared with two other ones from the literature, and shows better results in term of speed of convergence and overall results in the showed setups.

Considering the pros and cons detailed later, I think that the content is good but needs some integration in the experiments and above all a refactoring of figures and an integration with NL explanation of section 3

Pro:

  • The paper is well written
  • The underlying model is formally described in detail
  • The scenario faced is concrete and interesting, and the problem to be solved up to date and important
  • Results are promising
  • The overall approach is interesting and reusable

Cons:

  • The model presentation is really too formal, and it is quite incomprehensible for reading not really and fully skilled on these topics
  • The other two method selected for the comparison seem a subset of the proposed one (or, the proposed method is quite similar to the other two approaches with some more parts): it means that by construction the proposed approach should act better than the other two. I would have selected, if any, approaches from the literature which are similar to the presented one (that is, which consider all the aspects considered here, not a subset), otherwise they start directly “with an handicap”
  • The experiments setup is not so clear, and the results not so well described. That section needs a refactoring, and some more experiments with more agents should be provided (6 are not “many agents”, nor 30… 100 maybe?)
  • Sections “threat to validity” and/or a deep discussion is missing, as a real comparison with the state of the art too

Specific points

  • The introduction contains too detailed information for an introduction (which seems partially a “related work” section: this part should be moved in a dedicated section, and must be improved in details and above all in comparison with the presented approach), while misses an high level description of the presented method and of the results. In the introduction the reader, even one not completely of this specific research area, should be able to understand your general approach and the results
  • The formal description of the approach is really detailed, and not completely clear (for a “not of the area” reader is incomprehensible): even if it is ok that this section is clear for a skilled reader, and less for a not skilled one, it should not be completely unintelligible. I strongly suggest to add some “natural language” explanation of the overall approach, of the main concepts, and then of the main formulas, so that everyone can have at least a general idea of your method.
  • Related to the previous points, from the introduction and then going on, you should first of all introduce and explain the main concepts you are going to use, not only cite them and give them as “known”. “Attention-based methods”, “irrilevant agent”, “posterior distributions of variational auto-encoder”, “actor-critic framework” and so on are not always known. I suggest improving (or add a dedicated one) section 2 by explaining these concepts in a simple way, and to give an high level overview of your method, maybe introducing some simple explanation of the techniques you are going to exploit. You have some clear explanations, for example in lines 121-128 the description is clear, or the beginning of 3.3, but they come too late
  • Figures 1 to 5 really need a Legenda, and a better label explaining how to read the and interpret them
  • Formula (1) seems missing the explanation of some elements (E, pgreco), and in general it is not so clear to me the discount concept (discount wrt what…?). Could you please try to explain it better?
  • Parameter of table 1 need a description in the text, the general setup of the experiments is not so clear. In particular, I do not well understand the number of used agents, how they are incremented during the simulations and so on
  • How did you perform the simulation? Using which tool?
  • Please discuss why, in fig 7, maddpg performs better than your approach, and above all, I do not understand what is reported on x axis (and this is probably related to the missing explanation of table1 and of the experiment in general)
  • “the simulation results of 100 independent simulations for each scenario” means that you performed 100 run with 6 agents, 100 with 7…? or 100 in total? Please clarify better
  • Table 2 is unreadable, it needs rows and columns dividing the data, which are not simple to be linked with the lines. Furthermore, I suggest underlying in the tables the important results you want to show, to guide the reader in the understanding of the data. What is N? the number of agents? So, you used 29 agent maximum?
  • “t value of AERL” AERL…??
  • 3 does not describes the experiment in a clear way. For example, first it seems that you do not perform all the 3 methods but you refer to only two (why?), later it seems that you exploited all the 3, so, please include all of them from the beginning of the discussion
  • “predators”?
  • In fig 8, please give a name to the landmarks. Agents 2 and 5 are your choice to show some results? I deduce yes, but from the description it seems that they are the only two you can describe
  • If “Therefore, the number of SCN layers is set to 3 in this paper”, I assume you are using this in the experiments: so, why in Fig8 the two agents see only those two connected agents (they consider only them), and not all (with 3 steps you reach all the connected agents)? I would expect that the agents that are far get less attention, ok, but still some attention (while in fig8 they have 0)
  • Please improve the conclusion section, and add a discussion part somewhere of the problems and limitations of your approach, threat to validity and so on
  • “shows that the effectiveness” please remove “that”. “R collson” maybe is collision?
  • Please check the typos, there are some you should simply find

Reviewer 2 Report

The paper presents the proposition of multi-agent hierarchical policy for
agent cooperation. It deals with interesting and quite important topic. The paper is quite well prsented and organized, how it has some drawbacks.

The weak points of the paper are the following:

  1. no clear specified contribution of the paper
  2. there is no description of paper's structure in the Introduction
  3. each subsection should be started at least with the short introduction - it is a reserach paper, NOT the technical report
  4. the names of sections are not clear - they should be longer then 1 word
  5. figures are too small
  6. there is no disscussion of the experiments results
  7. conclusions are too short and general presented

Round 2

Reviewer 1 Report

Thanks for your detailed answer, and for having faced quite all my comments, I really appreciate it and I think that the paper really improved.

I only suggest to:

  • Add a list of acronyms at the end of the paper, to help the reader in remembering them (they are many)
  • Have a final check for typos (I see some in the new parts)
  • In the introduction, you still have a reference to AERL, please correct (and check in the rest of the paper please)
  • I think that your Response20 could help in the paper too to understand Figure8, please consider adding it (as it is or in a similar form, but the explanation with the matrix helps and makes clear the process)

Thanks again for your valuable work in updating the paper.

Reviewer 2 Report

The authors have addressed my comments and articulated their answer quite well. I have no further comments at this stage. I believe the paper is ready for publications.

Author Response

Thank you very much for your time and efforts spent on reviewing our manuscript.  We also highly appreciate your professional and valuable comments on our manuscript, and the suggestions are quite helpful for us to improve the quality of our paper.